# Pharmaceutical Prevention and Management of Cardiotoxicity in Hematological Malignancies

**DOI:** 10.3390/ph15081007

**Published:** 2022-08-16

**Authors:** Anastasia Stella Perpinia, Nikolaos Kadoglou, Maria Vardaka, Georgios Gkortzolidis, Apostolos Karavidas, Theodoros Marinakis, Chrysostomi Papachrysostomou, Panagiotis Makaronis, Charikleia Vlachou, Marina Mantzourani, Dimitrios Farmakis, Konstantinos Konstantopoulos

**Affiliations:** 1Department of Cardiology, “G. Gennimatas” General Hospital, 11527 Athens, Greece; 2Medical School, University of Cyprus, Nicosia 2029, Cyprus; 3Department of Hematology, “G. Gennimatas” General Hospital, 11527 Athens, Greece; 4First Department of Internal Medicine, Medical School, National and Kapodistrian University of Athens, “Laiko” General Hospital, 11527 Athens, Greece; 5Department of Hematology, Medical School, National and Kapodistrian University of Athens, “Laiko” General Hospital, 11527 Athens, Greece

**Keywords:** leukemia, lymphoma, cardiotoxicity, heart failure, anthracyclines, anticancer targeted therapies

## Abstract

Modern treatment modalities in hematology have improved clinical outcomes of patients with hematological malignancies. Nevertheless, many new or conventional anticancer drugs affect the cardiovascular system, resulting in various cardiac disorders, including left ventricular dysfunction, heart failure, arterial hypertension, myocardial ischemia, cardiac rhythm disturbances, and QTc prolongation on electrocardiograms. As these complications may jeopardize the significantly improved outcome of modern anticancer therapies, it is crucial to become familiar with all aspects of cardiotoxicity and provide appropriate care promptly to these patients. In addition, established and new drugs contribute to primary and secondary cardiovascular diseases prevention. This review focuses on the clinical manifestations, preventive strategies, and pharmaceutical management of cardiotoxicity in patients with hematologic malignancies undergoing anticancer drug therapy or hematopoietic stem cell transplantation.

## 1. Introduction 

New treatment modalities in hematology have improved the prognosis of patients with hematological malignancies (HM). Between 2006 and 2016, incident cases of leukemia and non-Hodgkin lymphoma increased globally by 26% and 45%, respectively, because of population growth and ageing [1]. In Europe, age-standardized incidence rates were 24.5 (per 100.000) for lymphoid malignancies and 7.55 for myeloid malignancies [2]. The overall incidence was lower in women than men and lowest in Eastern Europe [2]. 

The survival of HM has improved over the past 15 years, according to a EUROCARE-5 study, predominantly driven by new anticancer drugs. A 10% increase in survival occurred even in elderly patients with specific types of HM [3]. Nevertheless, cardiovascular diseases (CVDs) remain a significant cause of morbidity and mortality in patients with HM. Most importantly, onco-hematology treatments exaggerate the risk for CVDs, and they triple the risk of cardiovascular events [4]. Children surviving HM have a 7-fold higher mortality rate, 10-fold increased rates of CVDs and a 15-fold higher risk of developing congestive heart failure (HF) than their siblings [5]. Hence, CVDs may jeopardize the improved outcomes of modern therapy in patients with HM.

Cardiotoxicity in HM results from the interaction of the following three main factors: anticancer therapy, background cardiovascular status and HM itself [6]. Anticancer therapy may cause direct or indirect injury affecting all components of the cardiovascular system depending on patients’ cardiovascular status, coexisting risk factors and CVD. Finally, cancer per se may also affect the CV system, mostly indirectly, thereby contributing to the cardiovascular morbidity and mortality. 

In the present literature review, we presented the available data regarding the pathogenesis of cardiotoxicity and the potential pharmaceutical preventive strategies in patients with HM. 

## 2. Cardiotoxicity from Systemic Anticancer Drugs

Traditional chemotherapy and targeted therapies are associated with an increased risk of left ventricular dysfunction (LVD), heart failure (HF), hypertension, vasospastic and thromboembolic ischemia, and rhythm abnormalities, involving conduction system impairment and possible QTc prolongation, which can be life-threatening in rare cases [7,8,9,10] (Table 1 and Table 2).

### 2.1. Cytotoxic Agents 

#### 2.1.1. Anthracyclines (ANT)-Induced Cardiotoxicity

Anthracyclines, namely doxorubicin, daunorubicin, epirubicin, idarubicin, and mitoxantrone treat lymphomas and leukemias. Unfortunately, these drugs have a cumulative dose relationship with cardiotoxicity. Soon after anthracycline exposure, cardiomyocyte damage may appear, expressed clinically as dysrhythmia, repolarization alterations, pericarditis, and myocarditis. This acute form occurs at the time or during the first week of administration and resembles acute toxic myocarditis [8]. The chronic form appears later and is characterized by myocardial dysfunction. A summary of all suggested underlying mechanisms is displayed in (Figure 1) [10,11,12,13,14].

HF is the most important manifestation of cardiotoxicity with dose-dependent incidence, ranging from 3–5% at 400 mg/m^2^ doxorubicin to 18–48% at 700 mg/m^2^ [9]. Given this magnitude of cardiotoxicity, the indicated typical cumulative dose decreased to 240–360 mg/m^2^ doxorubicin and 450–600 mg/m^2^ epirubicin, which are doses associated with a 2–3% risk of HF development within at least five years [15]. However, subclinical cardiomyopathy occurred in 28% of patients with non-Hodgkin’s lymphomas receiving doxorubicin-based chemotherapy [16]. Diastolic dysfunction can appear at lower cumulative doses of 200 mg/m^2^, preceding systolic dysfunction [17]. According to previously published studies, the overall prevalence of cardiotoxicity may vary from 11.1% up to 27.6% [18], and the concomitant administration of radiotherapy worsens anthracycline-induced-cardiotoxicity [19].

#### 2.1.2. Alkylating Agents

Cyclophosphamide, ifosfamide, melphalan and busulfan are fundamental components in the treatment of chronic myelogenous leukemias, Hodgkin’s and non-Hodgkin’s lymphomas and as initial therapy for bone marrow transplantation (high dose). The hepatic metabolism of cyclophosphamide leads to the formation of aldophosphamide, which in turn is degraded into phosphoramide mustard and acrolein. The latter is a toxic metabolite that affects the heart and endothelial cells. Cyclophosphamide is responsible for changing the energy pool of cardiomyocytes by modulating the cardiac fatty acid binding proteins (H-FABP) [20]. Cyclophosphamide-induced cardiotoxicity involves multiple mechanisms; oxidative stress, apoptosis, myocardial inflammation, endothelial dysfunction, calcium dysregulation, endoplasmic reticulum and mitochondrial damage and a reduction in ATP synthesis [20]. Previous treatment with anthracyclines or chest irradiation increases the risk of Cyclophosphamide-induced cardiotoxicity [19]. In case of ifosfamide, left ventricular dysfunction occurs at doses exceeding 12.5 g/m^2^/cycle [8,21], and myocardial ischemia, cardiac tamponade, and arrhythmias may also occur [22].

#### 2.1.3. Antimetabolites

Common antimetabolites in the treatment of HM are clofarabine and cytarabine. Clofarabine is used in patients with recurrent or refractory acute leukemia, and it was only recently included in first-line regimens in acute leukemia patients. Most expected expressions of cardiac adverse effects are left ventricular systolic dysfunction, arrhythmias, hypo-/hypertension, pericarditis, pericardial effusion, angina, and rarely myocardial infarction [8,23]. Clofarabine inhibits DNA synthesis and ribonucleotide reductase while promoting apoptosis [23]. 

#### 2.1.4. Vinca Alkaloids

Vincristine and vinblastine are used in the treatment of leukemias, lymphomas, as well as multiple myelomas. They can result in myocardial ischemia, infarction, and arrhythmias [8,24]. Their actions typically result in a halt in cell division with an arrest at the mitotic metaphase/anaphase junction, followed by death [24].

## 3. Molecular-Target Agents 

### 3.1. Tyrosine Kinase Inhibitors (TKIs)

TKIs are currently used in a wide spectrum of hematologic malignancies. They are classified into the following two main classes: (1) small molecules TKIs and (2) monoclonal tyrosine kinase antibodies [8,25].

#### 3.1.1. Small-Molecule TKIs 

Dasatinib, nilotinib and notably ponatinib are associated with a high incidence of arterial thrombosis, whereas vascular thrombosis is less often observed. There is also an increased rate of thrombotic events in patients treated with combination therapy for multiple myeloma [9,26].

Ponanitib is used in the treatment of chronic myeloid leukemia (CML) and Philadelphia chromosome-positive acute lymphoblastic leukemia (ALL), resistant to traditional TKIs [8]. The phase II PACE multi-center trial showed that 11% of patients developed arterial thrombosis, and the events were serious in 8% of patients [27].

Nilotinib is also associated with an increased risk of arterial thrombosis, notably in the arteries of lower limbs and renal and mesenteric arteries as well [28].

Apart from arterial thrombosis, dasatinib has been implicated in precipitating pulmonary arterial hypertension (PAH) in patients treated for CML. Increased PAH incidence was observed at 36 months in the follow-up of the DASISION study [25]. Clinical and functional improvement was observed following dasatinib withdrawal; however, some patients also required PAH-specific treatment [29,30]. Isolated events of HF, pericardial effusion, hypertension, QT interval prolongation, and arrhythmias have also been reported in patients treated with dasatinib [31].

Left ventricular dysfunction (incidence ≤ 1%), largely reversible, pericardial effusion, tamponade, peripheral oedema through a non-cardiogenic mechanism, QT interval prolongation, arrhythmias and hypertension are the main types of cardiotoxicities seen with imatinib [22]. 

Ibrutinib, approved for the treatment of mantle cell lymphoma and chronic lymphocytic leukemia, has been associated with supraventricular and ventricular tachyarrhythmias [32,33].

A recent study that aggregated data from 1505 ibrutinib-treated individuals found 6.5% and 10.4% incidence of atrial fibrillation (AF) at 16.6 months and 36-month follow-up, respectively. Ibrutinib treatment, prior history of AF, and age of over 65 years were independent risk factors [32].

#### 3.1.2. Monoclonal Tyrosine Kinase Antibodies

Rituximab is a chimeric monoclonal antibody used against the CD20 antigen, commonly used for non-Hodgkin’s lymphomas, in addition to doxorubicin-containing treatment, significantly decreasing the risk of death. Although increased cardiotoxicity has not been detected so far, rare cases of non-ischemic cardiomyopathy, coronary spasm and Takotsubo syndrome have been reported [34,35]. At regular doses, AF can occur during or shortly after infusion [33]. 

Alemtuzumab-associated cardiotoxicity is rare and includes hypo/hypertension and arrhythmias [8].

### 3.2. Proteasome Inhibitors

Bortezomib and carfilzomib are newer chemotherapy agents targeting the ubiquitin–proteasome pathway. The first compound is used for the treatment of non-Hodgkin’s lymphomas and multiple myeloma, whereas the second is licensed as second-line therapy for refractory or relapsed multiple myeloma [36]. 

In a meta-analysis of studies including patients with various malignancies, the use of bortezomib was associated with a 3.8% rate of cardiovascular events. However, randomized studies did not find a significantly increased risk of cardiotoxicity compared to control patients [37].

On the contrary, carfilzomib usage significantly increased the risk of cardiomyopathy, uncontrolled hypertension, arrhythmias, ischemic events, and cardiac arrest in a dose-dependent manner [38,39]. A meta-analysis of 24 prospective trials involving 2594 patients with multiple myeloma revealed a rate of 18.1% of all-grade cardiovascular adverse events including symptomatic acute coronary syndrome, valvular disease, tachyarrhythmias, left or right ventricular dysfunction, and death [38,39].

### 3.3. Immune Checkpoint Inhibitors (ICPIs) 

ICPIs are a novel type of cancer treatment that boosts T-cell-mediated immune responses against cancer cells. Nivolumab and pembrolizumab, targeting the programmed death-1 receptor (PD-1), are licensed for classical Hodgkin disease treatment [21]. Rare but serious cardiotoxicity includes myocarditis, pericarditis, cardiac arrest, and HF [40]. ICPI-associated myocarditis often develops within two months of starting medication, but cardiotoxicity can occur at any point during or even after treatment has ended due to chronic immunological dysfunction. The incidence of fulminant cases of ICPIs-related myocarditis was reported as 1.14% in a recent multicenter registry [41], but with a high fatality rate reaching 50% [42]. Troponin levels were shown to be raised in 10% of patients receiving nivolumab treatment, suggesting that the real incidence of subclinical or smoldering myocarditis may be substantially greater [43]. Concomitantly diabetes and obesity showed independently high occurrence [43].

### 3.4. Chimeric Antigen Receptor T Therapy (CAR-T)

CAR-T cells use the immune system to kill cancer cells. They are patient-derived T-cells modified to target antigens on cancer cells, such as CD19 [22]. CAR-T cells revolutionized the treatment of hematologic malignancies, including lymphoma and leukemia as they were associated with significant response rates of up to 94% [44]. 

Cytokine release syndrome (CRS), which is clinically characterized by flu-like symptoms, circulatory shock, and potentially multi-organ failure, is one of the most prevalent and noticeable adverse effects of CAR-T therapy. Cardiotoxicity, secondary to CRS, includes arrhythmias, HF, and cardiovascular-related fatalities [22], ranging from 5.8 to 10.3% [45,46].

## 4. Hematopoietic Stem Cell Transplantation

Hematopoietic stem cell transplantation (HSCT) may be a cure for different HM. Compared to matched cohorts, survivors of HSCT are at increased risk of cardiovascular events or death. Cardiotoxicity may occur during HSCT as acute, including HF, arrhythmias, pericardial tamponade, or cardiac arrest, or late complication including cardiomyopathy, ischemic heart disease, vascular disease, and stroke [47]. Long-term survivors encounter a risk for CVDs at least four times higher than the general population [48].

HSCT consists of myelo-suppressive chemotherapy with or without total body irradiation accompanied by hematopoietic stem cell graft infusion. Different protocols are used, often involving irradiation and other agents [47,49]. Cardiovascular problems are also linked to several other variables, including the patient’s age, patients’ co-morbidities, cardiotoxic chemotherapy prior to HSCT, and the kind of HSCT (allogeneic vs. autologous) [50]. Excessive iron accumulation from transfusions causes cardiomyopathy by generating free radicals [51]. 

### 4.1. Acute Cardiotoxicity

It can be expressed as HF, arterial events, tamponade, and rhythm disturbances. One retrospective study by Tonorezos et al. reviewed 1177 adult patients receiving allogeneic and autologous HSCT and found that 5.3% of patients had post-transplant arrhythmias, associated with increased in-hospital and 1-year mortality [52]. Arrhythmias, mostly observed in patients with lymphomas, were associated with longer hospitalization, an increased probability of needing intensive care and a greater risk of death within one year after transplant [52]. In a more recent, single-center, retrospective study with 669 patients undergoing allogeneic transplantation, arrhythmias, congestive HF and myocardial infarction were observed in 0.9%, 2.2% and 0.9% of patients, respectively [53].

### 4.2. Late Cardiotoxicity

HSCT survivors remain at an increased risk for cardiovascular complications many years after transplantation. Relative to a general population, the risk of late death because of cardiotoxicity was four times higher in females after autologous HSCT and 2.3-fold higher in males and females after allogeneic HSCT [50]. In addition to sex, older age, hypertension, diabetes mellitus, and previous exposure to anthracyclines at dosages of more than 250 mg/m^2^ were all found to be synergistic risk factors for heart failure [52]. Armenian et al. showed that the incidence of HF in 1244 autologous HSCT survivors was 4.8% at five years and 9.1% at 15 years after transplantation [54].

Post-HSCT patients are also at increased risk of arterial events, including coronary artery disease, stroke, and peripheral artery disease, especially in the case of allogeneic transplantation [47], in which GVHD was implicated as a mechanism leading to arterial disease [55].

## 5. Prevention and Management of LV Dysfunction

### 5.1. Primordial Prevention

Its main target is to impede the occurrence of cardiotoxicity-associated risk factors by educating patients and providers and by implementing best practices and guidelines before cardiotoxicity occurs [56].

### 5.2. Primary Prevention-Preclinical Trials

Anthracycline compounds are the primary cause of chemotherapy-induced cardiotoxicity in patients with hematologic malignancies. Anthracycline compounds and new anticancer therapies, such as TKIs and immune-based therapies, were initially tested for cardiotoxicity using cultured cell and organoid models or in vivo models of small and large animals. Unfortunately, all those models cannot accurately represent the complicated process of cardiotoxicity in cancer patients. Indeed, cancer pathophysiology is complex, and animal models cannot mimic all the patients’ comorbidities. Furthermore, physiology, drug metabolism and gene expression differ between animal models and human beings [57,58]. Therefore, preliminary experimental studies did not fully provide the whole spectrum of the potential cardiotoxic effects of novel anti-cancer therapy.

Notably, in the case of ponatinib, preclinical testing in animal models and early clinical studies failed to detect cardiotoxicity; however, drug administration resulted in unwelcome cardiac side effects in HMs patients. Additional data from multiple ponatinib studies have shown a significant rate of major adverse cardiac events, outlining the shortcomings of current preclinical models. The reasons for those discrepancies are the different dosing and the shorter follow-up period [59]. More work is needed to elucidate the long-term adverse cardiac effects of ponatinib and evaluate preventive tailored treatment [60].

Regarding cardioprotective therapy, there were a number of preclinical trials using animal models which reported equivalent preventive effects of angiotensin-converting enzyme inhibitors (ACEis), angiotensin II receptor blockers (ARBs) or statins [61] against doxorubicin-induced cardiomyopathy.

### 5.3. Primary Prevention-Clinical Trials

Primary prevention includes limiting anthracycline dose, using novel anthracycline analogues (epirubicin instead of doxorubicin), or liposomal formulations, altering anthracycline administration, administering dexrazoxane, in conjunction with anthracycline treatment, and adding agents to halt cardiovascular dysfunction [62]. Dexrazoxane and cardiovascular agents appear to show similar efficacy in decreasing cardiotoxicity [63]. Overall, there is a lack of robust evidence for the primary prevention of cardiotoxicity. Ongoing numerous clinical trials aim to assess the impact of HF established therapy in the primary prevention of cardiotoxicity (Table 3).

#### 5.3.1. Dexrazoxane

The proposed beneficial effects of dexrazoxane are based mainly on two fundamental mechanisms. Firstly, dexrazoxane is an iron-chelating agent that within cardiomyocytes rapidly converts to its active form and counteracts the formation of anthracycline-iron complexes and the subsequent formation of reactive oxygen species (ROS). Secondly, dexrazoxane changes Top2beta’s structure, blocking its interaction with anthracycline and inhibiting the formation of Top-DNA cleavage complexes (Table 4). So far, it is the only FDA-approved drug for anthracycline-induced cardiotoxicity [64]. In a recent meta-analysis, patients co-treated with dexrazoxane had a 1/3 lower risk of HF compared to patients treated with only anthracycline, with a similar survival rate and incidence of secondary cancers [65]. Although most studies were carried out on non-HM, previous and recent data have shown that it may exert protective effects on hematological patients as well [66,67].

#### 5.3.2. Beta-Blockers

Data from clinical trials suggest a cardioprotective role for these agents in patients receiving chemotherapy [68] (Table 3). The cardioprotective effects of new-generation b-blockers, e.g., carvedilol, have been studied. Carvedilol, a nonselective b- and a1-adrenergic receptor (AR) antagonist, with its high antioxidant capacity, demonstrated improved resistance to ANT-induced cardiomyopathy.

The protective effect of carvedilol appears to be associated with its antioxidant capabilities rather than the b-AR obstructive activity and the consequent suppression of catecholamines. Carvedilol can protect against ANT-induced ROS production, endothelial dysfunction, cardiomyocyte apoptosis, alterations in mitochondrial respiration, and calcium overloading, resulting in improved lusitropy (cardiac relaxation) [64,69,70] (Table 4).

Carvedilol protected against adriamycin-induced LV dysfunction in children with acute lymphoblastic leukemia, preserving fractional shortening and global longitudinal strain (GLS) [71]. Similarly, carvedilol preserved LV diastolic and systolic function in adult patients receiving anthracycline [72,73,74]. In the OVERCOME trial, in which only patients with HM were evaluated, carvedilol in combination with enalapril prevented LVEF reduction compared to control group during the 6-month follow-up [75].

In a meta-analysis evaluating the efficacy of carvedilol for the primary prevention of anthracycline-induced cardiotoxicity in patients with breast cancer or hematological malignancies, it was found that prophylactic administration reduced the early onset of LV dysfunction compared with placebo [76]. In a recent systematic review and meta-analysis, the prophylactic use of carvedilol attenuated the frequency of clinically overt cardiotoxicity and prevented ventricular remodeling [77].

#### 5.3.3. ACEis and ARBs

The fundamental mechanisms of anti-anthracycline-related cardiotoxicity of ACEis and ARBs contribute to neutralizing ROS damage, lowering interstitial fibrosis and inflammation, preventing intracellular calcium excess, and improving mitochondrial respiration and cardiomyocyte metabolism [69,78] (Table 4).

In this context, enalapril further prevented systolic and diastolic dysfunction and decreased markers of myocardial injury [79]. Valsartan, administered simultaneously with anthracycline treatment, also prevented the increase in left ventricular diastolic diameter, natriuretic peptides levels and QTc interval duration [80]. Telmisartan, studied in patients treated with epirubicin for various cancers, demonstrated no impairment in myocardial deformation parameters [81]. It was also found that telmisartan could reverse acute epirubicin-induced myocardial dysfunction and maintain a normal systolic function up to the 12-month follow-up [82].

In contrast, other studies evaluating the protective role of enalapril or metoprolol in doxorubicin-induced cardiotoxicity, failed to demonstrate any favorable result [83]. Furthermore, in a recent meta-analysis examining adult patients undergoing chemotherapy and neurohormonal therapies, a significant heterogeneity in the treatment effects was observed, whilst the absolute gain in LVEF was modest, promoting the need for larger trials [84].

#### 5.3.4. Statins

Statins exert their protective action by inhibiting Top2β-mediated DNA damage in addition to pleiotropic antioxidant, antifibrotic and anti-inflammatory properties [85] (Table 4). In patients without preexisting cardiovascular abnormalities, the administration of 40 mg atorvastatin daily led to the preservation of LVEF [86]. It was also shown in a retrospective study that individuals receiving statin therapy for CVD prevention, may experience less deterioration in LVEF than individuals not receiving statins [87].

### 5.4. Secondary Prevention

Secondary prevention refers to the detection of at least subclinical cardiotoxicity and proactive cardio-protective therapy to inhibit or preferably reverse the deterioration of cardiovascular dysfunction [88].

Among 114 cancer patients with early troponin increase one month after chemotherapy initiation, enalapril was administered in 58 patients and continued for one year. Enalapril-treated patients did not develop LV dysfunction vs. 43% in the control group [89]. A multicenter randomized trial compared two strategies for guiding prevention with enalapril. In the prevention arm, all patients received enalapril at the beginning of their first chemotherapy, whereas in the troponin-triggered arm, enalapril was administered only after increased troponin. No differences were observed between preventive or troponin-triggered enalapril-based strategies, suggesting that in patients without heart disease receiving low doses of anthracyclines, enalapril therapy may be protective independent of a troponin-triggered strategy [90].

In terms of anthracycline-induced left ventricular dysfunction and HF, Cardinale et al. studied 201 consecutive patients with LVEF ≤ 45% due to anthracycline chemotherapy [91]. Enalapril and, when possible, carvedilol was initiated after the detection of LVEF impairment; overall, 42% of patients had full LVEF recovery, 13% had partial recovery, and 45% were non-responders. As the time from the end of chemotherapy to the beginning of HF treatment increased, the percentage of respondents gradually decreased; no complete recovery of LVEF was seen if HF treatment had been initiated after six months [91]. These promising results were confirmed by a larger study on 2625 patients receiving anthracycline-based chemotherapy mainly for breast cancer or non-Hodgkin’s lymphoma. Close monitoring of LVEF in this study allowed for the early detection of cardiotoxicity and prompt treatment with enalapril and carvedilol or bisoprolol, which resulted in partial recovery of LVEF in 71% of cases, though only 11% had a full recovery [92].

In a retrospective multicenter registry including patients with HF and HM, treated with sacubitril/valsartan in addition to conventional therapy N-terminal pro-B-type natriuretic peptide levels, functional class, and LVEF improved at the end of follow-up [93].

**Table 3 pharmaceuticals-15-01007-t003:** Trials evaluating preventive strategies with cardiovascular drugs in patients with hematologic malignancies.

First Author/Year	Chemotherapy	Design Medication	Results
Bosch et al. [75]	Anthracycline	Enalapril + Carvedilol vs. control (no treatment)	Treatment: LV ejection fraction (LVEF) −0.17%, Control: LVEF −3.28%, (*p* = 0.04)
Georgakopoulos et al. [83]	Anthracycline	1:1:1: Metoprolol, Enalapril. Control (no treatment	Metoprolol: HF 2%, enalapril HF: 5%, Control: 8% (*p* = 0.05)
Cardinale et al. [89]	Multiple	Enalapril vs. Control	Treatment: LVEF ↓ in 43%, Control: LVEF ↓ in 0%, (*p* < 0.001)
Cardinale et al. [91]	Anthracycline	Enalapril alone or Enalapril+ Carvedilol	LVEF recovery: 42% responders, 13% partial, 45% non-responders. No complete LVEF recovery after 6 M. ↓, rate of cumulative cardiac events in responders (*p* < 0.001)
Cardinale et al. [92]	Anthracycline	Enalapril alone or Enalapril and b-blockers in case of cardiotoxicity	11% of patients had full recovery, and (71%) had partial recovery. Early detection and prompt therapy predict substantial recovery of cardiac function.
Janbadai et al. [79]	Anthracycline	Enalapril vs. Control	TnI and CK-MB levels were significantly higher in the control group. Enalapril preserved systolic and diastolic function.
Jhorawat et al. [73]	Adriamycin	Carvedilol vs. no treatment	In carvedilol group, EF remained unchanged, vs. control group, *p* < 0.05.
Salehi et al. [74]	Anthracycline	Placebo vs. carvedilol 12.5 mg, vs. Carvedilol 25 mg	Carvedilol protects diastolic function at a dose of 12.5, and both systolic and diastolic function at 25 mg.
Cadeddu et al. [81]	Epirubicin	Telmisartan vs. Placebo	Tissue Doppler strain rate normalized only in Telmisartan group at >300 mg/m^2^ epirubicin
Kalay et al. [72]	Anthracycline	Carvedilol vs. Control	Treatment: LVEF 70.5 → 69.7% Control: LVEF 68.9 → 52.3% (*p* < 0.001)
Nakamee et al. [80]	Anthracycline	Valsartan vs. Control	Valsartan prevented ↑ in LV end-diastolic dimension, demonstrated in the control group
Dessi et al. [82]	Epirubicin	Telmisartan vs. Placebo	TEL maintains a normal systolic function up to the 12-month FU.
El-Shitany et al. [71]	Adriamycin	Adriamycin vs. Adriamycin + Carvedilol pretreatment	Prevention of ↓ of Fractional shortening, Global Peak Systolic Strain; ↑ troponin.
Cardinale et al. [90]	Epirubicin or Doxorubicin in patients with low cardiovascular risk	Enalapril before chemotherapy or enalapril only in increased troponin	No differences were observed between preventive or troponin-triggered enalapril-based strategy
Martín-Garcia et al. [93]	70% anthracyclines 30% radiotherapy	Sacubitril/valsartan	↑ LVEF, ↓ LV diameters, ↓ NYHA class vs. placebo
Acar et al. [86]	Anthracycline	Atorvastatin vs. Control	Treatment: LVEF +1.3%; Control: LVEF −7.9%. (*p* < 0.001)
Chotenimitkhun et al. [87]	Anthracycline	Statin vs. Control	Treatment: LVEF –1.1%; Control: LVEF −6.5%. (*p* = 0.03)

**Table 4 pharmaceuticals-15-01007-t004:** Drugs protective against cardiotoxicity induced by anthracyclines.

Treatment	Mechanism
Dexrazoxane [64]	Reactive oxygen species reduction, prevention of cardiac Top2β anthracyclines interaction, reduction in DNA damage
ACEIs, ARBs [64]	Reduction of oxidative stress, antifibrotic and inti-inflammatory effects, improvement of intracellular calcium handling, mitochondrial function, and cardiomyocyte metabolism
Beta-blockers [64,69,70]	Reduction of oxidative stress and cardiomyocyte apoptosis, enhanced lusitropy, prevention of endothelial dysfunction
Statins [85]	Inhibition of Top2β-mediated DNA damage, anti-inflammatory, and antioxidant effects, reduction in myocardial fibrosis
Valsartan/sacubitril [93]	Reduction in myocardial fibrosis

In general, when cardiac dysfunction is established, the main goal is to limit patients’ disability and improve their quality of life. In most cases, anticancer therapy needs to be discontinued, at least temporarily, but prompt cardioprotective therapy may keep anti-cancer therapy on the tracks and avoid permanent discontinuation. The management of CVDs due to anticancer therapy follows the general strategies applied to patients without cancer. Additional parameters that need to be taken under consideration include potential interactions with anti-cancer therapy, life expectancy and cancer-related comorbidities [94]. Detailed guidance on the management of specific complications is provided by international scientific statements [62].

## 6. Prevention of Other Types of Cardiotoxicities

The screening process for ischemic heart disease in patients with HM does not differ from those without cancer. Because CVD results from risk factors accumulating over time, patients at high risk should be closely evaluated and cardiovascular risk factors aggressively treated. Similarly, the management of anti-cancer therapy induced hypertension or pericardial syndromes likening that of the general population [4,9,95].

## 7. Cardiotoxicity and Arrhythmia

### 7.1. Atrial Fibrillation (AF)

Patients with cancer may have a greater risk of AF than those without cancer. Newly onset AF in cancer patients might indicate a more advanced malignancy stage and worse oncological outcome. Cancer is expected to raise the risk of venous thromboembolism by 4 to 7 times, as well as a 2-fold increased risk of bleeding during initiation of anticoagulation treatment [96]. On a pathophysiological level, AF and cancer may interact with one another. There are various possible causes for this interconnection. Firstly, cancer-related systemic inflammation predisposes patients to atrial remodeling. The instability of the autonomous nerve system caused by pain and stress is a second hypothesized explanation for driving AF in cancer patients. Other possible causes are metabolic and electrolyte problems, fluid imbalance (during chemotherapy), and infections [94]. Finally, many anti-cancer treatments such as anthracycline agents, alkylating agents, and new targeted therapies, especially ibrutinib, may stimulate AF [97]. Moreover, even after many years, cancer patients are at increased risk of AF, with a hazard ratio of 2.47 (95% CI 1.57–3.88) [98].

In relation to management, β-blockers, such as atenolol or metoprolol, should be used as first-line agents. Depending on cardiovascular comorbidities, flecainide, propafenone, and sotalol may be alternative options for anti-arrhythmic medication treatment. On the other hand, amiodarone and the calcium-channel blockers diltiazem and verapamil, which are common drugs in cancer-free individuals, can boost ibrutinib plasma levels several-fold by interfering with its hepatic metabolism. As a result, they should be used with caution as substrates of P-glycoprotein, which is inhibited by ibrutinib. More than two-thirds of ibrutinib-treated AF patients who are planning to undergo cardioversion need to be treated with anti-arrhythmic medication as well [32].

### 7.2. QTc Interval Prolongation and Ventricular Tachycardia (VT)

Most ventricular arrhythmias caused by anti-cancer drugs are linked to QTc prolongation. A QTc of >500 ms, and a ΔQTc (change from baseline) >60 ms are both regarded as concerning [33]. A QTc > 550 ms is linked to a 2 to 3 times higher risk of torsades de pointes (TdP). Cancer patients appear to be particularly vulnerable, as both the disease itself and anti-cancer medications can produce hypokalemia (through vomiting and diarrhea). In addition, cancer patients frequently have co-morbidities that enhance the likelihood of QTc prolongation (like hypocalcemia or hypomagnesemia) [33].

Drugs used in HM that can result in VT with or without QTc prolongation include anthracyclines, arsenic trioxide, and TKIs such as imatinib, nilotinib, ponatinib, dasatinib, and ibrutinib [33]. After arsenic trioxide administration, one-third of patients experience a QTc prolongation of 30–60 ms from baseline and another third a prolongation of >60 ms, with 65% of patients with QTc > 500 ms. Torsades de pointes are seldom detected unless there are other contributory variables present, such as electrolyte disorders. Although sudden cardiac death has been documented, it is extremely uncommon [32]. For instance, QTc prolongations of 500 ms were observed in 5% of patients on TKIs, such as dasatinib, and nilotinib, but ventricular arrhythmia and sudden cardiac death were observed in <1% of them [99]. Results from registry-based research released in 2018 validated previous observations of ibrutinib’s risk of ventricular arrhythmias. The link between ibrutinib and ventricular tachycardias was regarded as at least plausible, and generally, a rate of VT that was more than ten times greater than predicted was found. Even in the context of a normal QTc, VT and ventricular fibrillation, including polymorphic VT, have been documented with ibrutinib administration [100].

Ventricular arrhythmias may also be caused by inflammatory infiltration of the myocardium in people using ICIs. Ventricular arrhythmias occur in 5–10% of individuals using ICIs and are connected to a 40% mortality rate. Ventricular arrhythmias, a signal of a more complex clinical history, should be investigated for the existence of myocarditis. Other types of cardiomyopathies that have been reported with ICI treatment might potentially cause ventricular arrhythmias [101].

In all patients starting anti-cancer treatment that prolongs the QTc interval, a baseline ECG should be acquired. Prior to starting therapy, electrolyte imbalances must be rectified, and drug–drug interactions that affect the QTc interval (antibiotics, anti-emetic, anti-psychotic, or anti-depressant medicines) must be examined [102].

The most suitable method for modulating the arrhythmogenic substrate and improving outcomes in patients with cancer therapy-induced arrhythmias is likely to be early detection followed by adequate care of cardiac ischemia, dysfunction, and remodeling. These guidelines apply to QTc prolongation and associated ventricular arrhythmias [32]. After dosage reductions or therapeutic breaks of more than two weeks, the same level of monitoring is necessary [32].

## 8. Prevention of Vein Thromboembolism (VTE)

Cancer and AF share common pathways of hypercoagulability. On top of that, it is also well established that certain types of treatments boost thrombotic risk [96]. Importantly, the CHA2DS2-VASc score used to choose the anticoagulation strategy does not account for cancer-induced hypercoagulability and performs poorly in patients who have recently developed AF [103]. On the other hand, the bleeding risk is also increased due to thrombocytopenia, as frequently seen in HM or after certain chemotherapies. Regarding bleeding risk prediction, differences in patients with cancer are also not included in the HAS-BLED score, and, for this reason, this score might not perform ideally in these patients [32].

Due to the likelihood of interactions between anticoagulant treatment and chemotherapy, increased awareness is required throughout therapy because unadjusted dose and treatment might lead to serious side effects, with thromboembolic and bleeding problems being more likely [104]. Although new oral anticoagulants (NOACs) do not require close monitoring, it is important to be aware of the potential for drug–drug interactions [96]. The European Heart Rhythm Association recently published a practical guide with several recommendations for the dosing and use of NOACs when used in conjunction with chemotherapy that induces or inhibits the permeability glycoprotein (P-gp) or cytochrome P450 (CYP3A4) systems, which are important in the elimination pathways of NOAC [105].

Four randomized trials investigated the efficacy and safety issues of NOACs versus low molecular-weight heparins (LMWH) in patients with cancer, and those with HM, albeit they were the minority.

In the HOKUSAI trial, which was an open-label, noninferiority trial, patients with cancer with acute symptomatic or incidental venous thromboembolism were randomly assigned to receive either LMWH for at least five days followed by oral edoxaban at a dose of 60 mg once daily (edoxaban group) or subcutaneous dalteparin at a dose of 200 IU per kilogram of body weight once daily for one month followed by dalteparin at a dose of 150 IU/kilogram once daily (dalteparin group). Treatment lasted at least 6 months and up to 12 months. The main outcome, regardless of treatment duration, was a composite of recurrent venous thromboembolism or significant bleeding within 12 months of randomization. Edoxaban had a lower incidence of recurrent VTE than dalteparin, although it had a greater rate of severe hemorrhage [106].

In the ADAM-VTE trial, the primary goal was to prevent significant hemorrhage. VTE recurrence and a combination of major plus clinically significant nonmajor bleeding were the secondary outcomes. Patients with cancer related VTE were administered subcutaneous dalteparin (200 IU/kg for one month followed by 150 IU/kg once daily) or apixaban 10 mg twice daily for seven days followed by 5 mg twice daily for six months. Overall, 66% of the participants had a metastatic illness, and 74% underwent chemotherapy at the same time. Apixaban arm group appeared with lower VTE recurrence than dalteparin, while no significant bleeding was reported [107].

Patients with active malignancy who had symptomatic pulmonary or incidental embolism or symptomatic lower-extremity proximal deep vein thrombosis (DVT) were enrolled in the randomized trial SELECT-D. Dalteparin (200 IU/kg) was administered daily during the first month, then 150 IU/kg twice daily for the next two months, or rivaroxaban (15 mg twice daily for three weeks and then 20 mg daily for six months. VTE recurrence during a 6-month period was the main outcome. Major hemorrhage and clinically relevant nonmajor bleeding were used to determine safety. Rivaroxaban was associated with relatively low VTE recurrence, and a higher rate of bleeding correlated to dalteparin [108].

In the CARAVAGGIO trial, patients with cancer who had symptomatic or incidental acute proximal DVT or pulmonary embolism were randomly assigned to receive oral apixaban at a dose of 10 mg twice daily for the first seven days, followed by 5 mg twice daily, or subcutaneous dalteparin at a dose of 200 IU/kg of body weight once daily for the first month, followed by 150 IU/kg once daily for six months. Recurrent VTE was comparable in both groups (*p* < 0.001 for noninferiority). Major bleeding occurred in 3.8% of patients in the apixaban group and in 4.0% of the dalteparin group, a difference that was non-significant statistically [109].

Overall, three out of four RCTs of cancer populations with VTE and some of them with HM, demonstrated the superiority of NOACs over dalteparin in VTE recurrence, with an equivalent bleeding risk. This is of clinical relevance for future therapeutic algorithms.

## 9. ICPIs Myocarditis

At present, ICPIs-related myocarditis is difficult to detect, even with continuous monitoring of cardiac troponins levels [110]. A sensitive cardiac troponin test can expose subtle changes in the heart. Further investigation is needed with other biomarkers analysis, electrocardiogram, and diagnostic imaging such as an echocardiogram to evaluate cardiac function, cardiac MRI for increased T2-signal intensity and abnormal early and late gadolinium enhancement and tagged cine MRI for strain analysis.

Cardiac MRI outperforms echocardiography in terms of identifying myocardial edema, inflammation, and fibrosis in the tissue. Because ICPIs-related myocarditis can cause immunotherapy to be stopped, tissue diagnosis from an endomyocardial biopsy is still advised if the other diagnostic tests are inconclusive [42]. In addition, GLS may be capable of detecting myocarditis in cases with either a reduced or preserved LVEF and to predict major adverse cardiac events regardless of LVEF [111].

Patients with ICPIs myocarditis may need to be admitted to the hospital, depending on clinical status, especially if hemodynamic instability or any substantial abnormalities are detected upon lab tests. The first-line treatment is a high dose of corticosteroids, often methylprednisolone at up to 1,000 mg/day for three days, followed by prednisone 1 mg/kg [112]. The greater the first dosage and the quicker the corticosteroid administration (within 24 h of presentation), the better the achieved results. The duration of treatment should be individualized, depending on the patient’s responsiveness, and during recovery, it should be tapered slowly after previous abnormal testing returns to normal [112].

For stable patients non-responsive to steroids, infliximab and mycophenolate mofetil might be considered if the biopsy reveals evidence of high-grade myocarditis [113]. Antithymocyte globulin, intravenous immunoglobulin, and plasma exchange should all be the second-line therapy if the patient is unstable [43].

Conventional cardiac therapies are also used in accordance with international recommendations. The American Society of Clinical Oncology recently issued a practice guideline where the discontinuation of ICPIs was recommended as soon as abnormalities in cardiac biomarkers or ECG are observed [114]. However, the decision may be considered for patients with a mild course and complete recovery from myocarditis [43].

## 10. Cardiotoxicity from Radiotherapy

Thoracic radiation therapy is a successful treatment for several hematological malignancies, such as Hodgkin’s lymphoma. The association between radiotherapy (RT) and cardiac dysfunction is well known and notably the RT-associated morbidity and mortality can diminish the increased life expectancy of anti-cancer therapies. Radiation-induced heart disease (RIHD) is characterized by a complex pathogenetic mechanism and covers a wide range of pathologies, including pericardial disease, cardiomyopathy, coronary artery disease, valvular disease, and arrhythmias [115]. Although radiation cardiotoxicity is beyond the scope of this review, it is worth mentioning possible protective measures.

The most valuable measure for prevention is to provide radiotherapy only to patients in whom it is required and at adequate doses; the lowest efficient dose should be administered [116]. Besides a close follow-up, secondary prevention incorporates radiomitigation approaches. So far, our current knowledge highlights several possible strategies that could forestall RIHD, with available cardiovascular drugs or new agents targeting the main pathogenetic processes [117].

### 10.1. Statins

Statins were shown to have several beneficial effects in preclinical studies. Lovastatin, and atorvastatin, have been found to reduce radiation-related deleterious effects by exerting anti-inflammatory and antifibrotic actions [117]. However, the recently published randomized, placebo-controlled pilot clinical trial, examining the effect of atorvastatin on vascular function and structure in young adult survivors of childhood cancer showed that six months of atorvastatin treatment did not ameliorate endothelial function or arterial stiffness [118].

### 10.2. ACEIs

ACEIs can lessen myocardial perivascular fibrosis and myocardial cell apoptosis through anti-inflammation and oxygen-free radicals production following simultaneous heart and lung exposure to radiation; in this way, they decreased myocardial fibrosis and diastolic cardiac dysfunction. Experimental studies presented promising results [117]. So far, prospective clinical studies evaluating the potential benefits of ACEIs to prevent RIHD are lacking.

### 10.3. Anti-Inflammatory and Antioxidants Agents

Anti-inflammatory agents, such as colchicine, dexamethasone, pioglitazone, and non-steroidal inflammatory drugs, as well as antioxidants agents such as pentoxifylline combined with α-tocopherol, melatonin and amifostine, decrease myocardial fibrosis and exert protective properties against RIHD [117,119].

### 10.4. New Compounds

Preclinical studies have also shown that glutathione S-transferase alpha 4 (GSTA4-4) inhibitors, sestrins activators, TGFβ receptor 1 inhibitors, and recombinant human neuregulin-1, through several underlying pathogenic mechanisms, preserve cardiac function and late myocardial fibrosis after radiation [120,121,122,123].

## 11. Conclusions

The prognosis of patients suffering from hematologic malignancies has been significantly improved. However, anti-cancer treatment induces various forms of cardiac adverse effects, increasing morbidity and limiting the survival of patients. Efforts have been made so far to minimize cardiac complications by using drugs with established cardiovascular protective properties adjusted in the cancer environment. Most studies recommend close monitoring using imaging modalities and biomarkers. In case of HM patients and concomitant CVD or those commencing anti-cancer therapy characterized by high cardiotoxicity risk (e.g., high-dose of anthracyclines), cardiologists should be proactive by prescribing cardioprotective drugs. In case of HM patients with even subtle cardiovascular dysfunction, established pharmaceutical agents should be promptly prescribed to reverse cardiovascular dysfunction. Data from randomized studies involving only patients with HM are rare; therefore, trials evaluating the prevention of cardiotoxicity in this heterogeneous group of cancer patients are required. The collaboration among hematologists, cardiologists and oncologists is of paramount importance in optimizing patients’ care.

## Figures and Tables

**Figure 1 pharmaceuticals-15-01007-f001:**
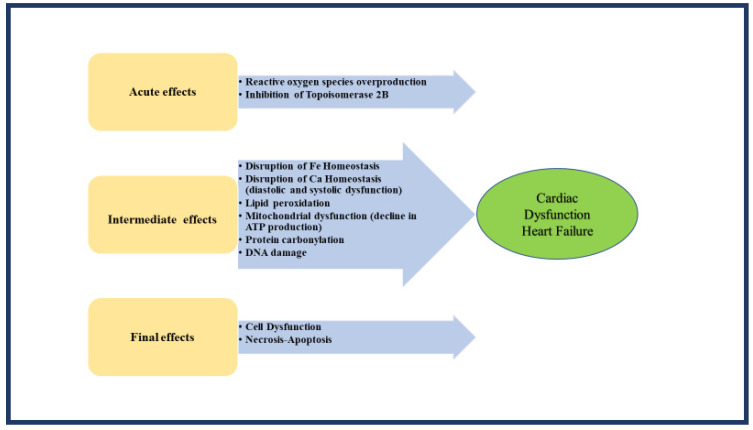
Mechanisms of Anthracycline-induced cardiotoxicity.

**Table 1 pharmaceuticals-15-01007-t001:** Chemotherapeutic agents with potential cardiotoxicity in hematologic malignancies.

Chemotherapeutic Class and Agents	Cardiomyopathy Incidence	Other Types of Cardiovascular Toxicity	Clinical Use in Hematologic Malignancies
Anthracyclines [7,8,9]			
Doxorubicin	(3–26)%	Myopericarditis, cardiac arrhythmias	Acute myeloid leukemia, acute myelogenous leukemia, chronic lymphocytic leukemia, Hodgkin and non-Hodgkin lymphoma
Idarubicin	(5–18)%	ECG abnormalities	Acute myeloid leukemia
Mitoxantrone	(0.2–30)%	Cardiac arrhythmias, ECG abnormalities	Acute nonlymphocytic leukemias
Alkylating agents [7,8,9]			
Cyclophosphamide (high dose)	(7–28)%	Peri-/myocarditis, cardiac tamponade, arrhythmias	Bone marrow transplant, chronic myelogenous leukemias
Ifosfamide	17%	Arrhythmias, cardiac arrest, myocardial hemorrhage, myocardial infarction	Hodgkin and non-Hodgkin lymphoma
Busulphan	Rare	Endomyocardial fibrosis, pericardial effusion, tamponade, ECG changes, chest pain, hyper-/hypotension, thrombosis, arrhythmias	Chronic myelogenous leukemia, hematopoietic stem cell conditioning regimen
Antimetabolites [8]			
Clofarabine	27%	Arrhythmias, hypo-/hypertension, pericarditis/pericardial effusion	Acute lymphocytic leukemia
Cytarabine	Undefined	Pericarditis, chest pain (including angina)	Hodgkin and non-Hodgkin lymphoma, acute leukemia (myeloid and lymphocytic)
Antimicrotubule agents [7,8]			
Vincristine	25%	Hyper-/hypotension, myocardial ischemia	Acute lymphocytic leukemia, Hodgkin and non-Hodgkin lymphoma, multiple myeloma
Monoclonal antibody-based tyrosine kinase inhibitors [8]			
Alemtuzumab	Rare	Hypo-/hypertension, arrhythmia	Chronic lymphocytic leukemia, cutaneous T-cell lymphoma, bone marrow transplant
Rituximab	Rare	Hypotension, arrhythmia	Non -Hodgkin lymphoma
Small-molecule tyrosine kinase inhibitors [7,8]			
Dasatinib	(2–4)%	Pericardial effusion, hypertension, arrhythmia, QT interval prolongation, Pulmonary arterial hypertension	Philadelphia chromosome + chronic myeloid leukemia and acute lymphoblastic leukemia
Imatinib mesylate	(0.5–1.7)%	Pericardial effusion and tamponade, anasarca, arrhythmias, hypertension, Raynaud disease	Philadelphia chromosome + chronic myeloid leukemia and acute lymphoblastic leukemia
Ponanitib	Undefined	Arterial thrombosis	Chronic myeloid leukemia and Philadelphia chromosome + acute lymphoblastic leukemia, resistant to traditional TKIs
Proteasome Inhibitors [7,8]			
Bortezomib	(2–5)%	Ischemia, bradycardia	Multiple myeloma, mantle cell lymphoma
Carfilzomib	8.68%	Uncontrolled hypertension	Relapsed or refractory multiple myeloma
Immune Checkpoint Inhibitors [7,10]			
Pembrolizumab	1%	Myocarditis, pericardial desease, conduction abnormalities	Hodgkin Lymphoma
Nivolumab	0.54%	Myocarditis, pericardial desease, conduction abnormalities	Hodgkin Lymphoma

**Table 2 pharmaceuticals-15-01007-t002:** Cardiotoxicity of the most common treatment protocols in different hematologic malignancies (according to National Cancer Comprehensive Network, www.nccn.org, accessed on 19 June 2022).

Type of Malignancy	Protocols	Cardiotoxicity
1. Hodgkin Lymphoma	1st line 1. ABVD: Doxorubicin, Bleomycin, Vinblastine, Dacarbazine 2. B + AVD: Brentuximab Vedotin, Doxorubicin, Vinblastine, Dacarbazine 3. BEACOPP escalated: Bleomycin, Etoposide, Doxorubicin, Cyclophosphamide, Vincristine, Procarbazine, Prednisolone	Heart Failure, heart attack, arrhythmias, pericardial effusion, peri-/myocarditis, hyper/hypotension, ischemia
2. Non-Hodgkin Lymphomas	1st line 1. R-CHOP: Rituximab, Cyclophosphamide, Vincristine, Prednisolone 2. R-da (dose adjusted) EPOCH: Rituximab, Etoposide, Vincristine, Doxorubicin, Cyclophosphamide, Prednisolone 3. CODOX-M/IVAC + R: Cyclophosphamide, Vincristine, Doxorubicin, High dose Methotrexate/ Ifosfamide, Etoposide, High Dose Cytarabine + Rituximab 4. R-Hyper—CVAD: Rituximab, Cyclophosphamide Vincristine, Doxorubicin, Dexamethasone/High Dose Methotrexate, High Dose Cytarabine 5. BR: Bendamustine, Rituximab	Arrhythmias, ischemia, heart failure, peri-/myocarditis, hyper/hypotension
3. Multiple Myeloma	1st line 1. VRd: Bortezomib, Lenalidomide, Dexamethazone 2. DRd: Daratumumab, Lenalidomide, Dexamethazone 3. CRd: Carfilzomib, Lenalidomide, Dexamethasone 4. VCd: Bortezomid, Cyclophosphamide, Dexamethazone 5. VAd: Bortezomib, Doxorubicin, Dexamethasone	Arrhythmias, atrial fibrillation, heart failure, myocarditis, hypertension
4. Acute Promyelocytic Leukemia	1st line 1. ATRA + Arsenic Trioxide (+/−Idarubicin, +/−Gemtuzumab Ozogamicin)	Pleural or pericardial effusion, Arterial and venous thrombosis, heart failure, QT prolongation
5. Acute Myeloid Leukemia	1. Idarubicin + Cytarabine (+/−Midostaurin, +/−Gemtuzumab Ozogamicin) 2. Azacitidine + Venetoclax 3. LDAC: Low dose Cytarabine + Venetoclax	Heart failure, arrhythmias, angina, pericarditis with effusion, QT prolongation, rarely edema, heart failure and myocardial infarction rarely induced by Venetoclax
6. Acute Lymphoblastic Leukemia	1. Hyper-CVAD +/− L-Asparaginase +/− TKI (Imatinibe, Dasatinib, Nilotinib, Bosutinib, Ponatinib) CVAD: Cyclophosphamide, vincristine, doxorubicin, dexamethasone	Heart failure and/or Myocarditis, pericardial effusion, hypertension, arrhythmia, QT interval prolongation, pulmonary arterial hypertension arrhythmias, hypertension, arterial thrombosis
7. Chronic Lymphocytic Leukemia	1. Ibrutinib 2. Venetoclax + Obinutuzumab 3. Bendamustine + Rituximab 4. FCR: Fludarabine, Cyclophosphamide, Rituximab	Atrial fibrillation, arrhythmias, hypotension, peri-/myocarditis

## Data Availability

Data sharing not applicable.

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
