# Peer review of "Pharmaceutical Prevention and Management of Cardiotoxicity in Hematological Malignancies"

_pharmaceuticals, 2022, doi:10.3390/ph15081007_

Round 1

Reviewer 1 Report

Interesting paper, minor language corrections are needed

Author Response

Dear Sir/Mrs

Thanks for your suggestions. Minor spell checking has been corrected.

Kind regards

Anastasia Perpinia

Reviewer 2 Report

The manuscript by Perpinia A et al describes the cardiotoxic manifestations caused by anti cancer drugs used in hematological malignancies. The review is well written and supported by tables and images. I have following comments for the manuscript:

1. Authors needs to write a paragraph on why these cardiotoxic adverse events were not reported in early clinical trials and preclinical studies.

2. A paragraph is required on need of improved preclinical studies for early detection of cardiac adverse events in developing drugs. Authors may cite these studies: PMID: 32470534, PMID: 26371140.

Author Response

Thanks in advance for your valuable suggestions. Two paragraphs are included about preclinical studies for early detection of cardiotoxicity in developing drugs. The first one refers to positive results in cardioprotection against doxorubicin-induced cardiotoxicity, and the second one is related to the discrepancies observed between preclinical and clinical studies in the case of ponatinib. In brief, I mention the reasons why these discrepancies exist. i have also entailed the proposed references. All the changes are highlighted in blue.

Kind regards

Reviewer 3 Report

Dear Authors,

I have read this review with interest. I also see it can be useful for practicing clinicians.

- please add citations to Tables

- the text needs some minor stylistic corrections (eg. too long spaces etc.)

- I would love to see some additional informations about molecular background of cardiac complications caused by systemic agents as well as molecular explanations of why these drugs might or might not work (one-two sentences more with each cardiotoxic AE or drug/groups will make it).

I wish you all the best with the paper.

Author Response

Dear Sir/Mrs

Thank you for your useful suggestions

I added the citations to the tables and corrected the stylistic errors. I have also added the molecular explanations of cardiotoxicity by systemic drugs. All the changes are highlighted in yellow.

Kind regards
